# Gelatin Methacryloyl Hydrogels for Musculoskeletal Tissue Regeneration

**DOI:** 10.3390/bioengineering9070332

**Published:** 2022-07-21

**Authors:** Yang-Hee Kim, Jonathan I. Dawson, Richard O. C. Oreffo, Yasuhiko Tabata, Dhiraj Kumar, Conrado Aparicio, Isha Mutreja

**Affiliations:** 1Bone and Joint Research Group, Centre for Human Development, Stem Cells & Regeneration, Institute of Developmental Sciences, University of Southampton, Southampton SO16 6YD, UK; jid@soton.ac.uk (J.I.D.); richard.oreffo@soton.ac.uk (R.O.C.O.); 2Department of Biomaterials, Field of Tissue Engineering, Institute for Frontier Medical Sciences, Kyoto University, Kyoto 606-8501, Japan; yasuhiko@infront.kyoto-u.ac.jp; 3Division of Pediatric Dentistry, School of Dentistry, University of Minnesota, Minneapolis, MN 55812, USA; kumard@umn.edu; 4Minnesota Dental Research Center for Biomaterials and Biomechanics, Department of Restorative Science, University of Minnesota, Minneapolis, MN 55455, USA; cjaparicio@uic.es; 5Division of Basic Research, Faculty of Odontology UIC Barcelona—Universitat Internacional de Catalunya, 08017 Barcelona, Spain; 6BIST—Barcelona Institute for Science and Technology, 08195 Barcelona, Spain

**Keywords:** gelatin, GelMA, hydrogel, drug delivery, musculoskeletal tissue

## Abstract

Musculoskeletal disorders are a significant burden on the global economy and public health. Hydrogels have significant potential for enhancing the repair of damaged and injured musculoskeletal tissues as cell or drug delivery systems. Hydrogels have unique physicochemical properties which make them promising platforms for controlling cell functions. Gelatin methacryloyl (GelMA) hydrogel in particular has been extensively investigated as a promising biomaterial due to its tuneable and beneficial properties and has been widely used in different biomedical applications. In this review, a detailed overview of GelMA synthesis, hydrogel design and applications in regenerative medicine is provided. After summarising recent progress in hydrogels more broadly, we highlight recent advances of GelMA hydrogels in the emerging fields of musculoskeletal drug delivery, involving therapeutic drugs (e.g., growth factors, antimicrobial molecules, immunomodulatory drugs and cells), delivery approaches (e.g., single-, dual-release system), and material design (e.g., addition of organic or inorganic materials, 3D printing). The review concludes with future perspectives and associated challenges for developing local drug delivery for musculoskeletal applications.

## 1. Introduction

Musculoskeletal tissue consists of muscle, bone, cartilage, tendon, ligament and other connective tissues, which comprise more than 40% of the human body by mass and support the shape and structure of the body. Musculoskeletal tissue is subject to wear and tear over a lifetime and can be subject to damage following traumatic injury and degenerative disease. Indeed, recent data on the global burden of disease showed that approximately 1.7 billion people globally suffer from musculoskeletal conditions and they are currently the highest contributor to the global need for rehabilitation [1]. To date, autograft, allograft and xenograft are used for musculoskeletal tissue treatment but they present a number of associated risks including immune rejection, infection and disease transmission, donor site morbidity and limited reproducibility [2]. 

Tissue engineering has emerged as a promising approach to treating musculoskeletal tissues through the incorporation of state-of-the-art multidisciplinary technologies involving material science, cellular and molecular biology and chemistry. Biomaterial-based scaffolds provide a microenvironment that can modulate cell behavior, including recruitment, adhesion, migration, proliferation and differentiation. In addition, bioactive molecules, such as drugs and growth factors, can be incorporated into the scaffold(s) to improve cell differentiation capacity to generate a target tissue. 

Among the plethora of scaffolds in development, hydrogels have attracted significant attention due to their tuneable physicochemical properties. Hydrogels are polymeric networks capable of holding a large amount of water that can be chemically crosslinked. Natural and synthetic polymer-based hydrogels have been widely used and chemically and physically modified to improve cell behavior and activate cell molecular signaling. In particular, natural polymers such as collagen, fibronectin, chitosan, hyaluronic acid, laminin and elastin are notable as biomimetic materials given their comparable composition to components of extracellular matrix (ECM). The natural polymer-based hydrogels display excellent biocompatibility, biodegradability, biological activity and low cytotoxicity, compared with synthetic polymers [3]. However, the weak mechanical properties and/or rapid degradation rate of natural polymers create challenges for clinical translation. Therefore, natural polymer-based hydrogels are often combined with synthetic polymers or ceramics, or in subsequent crosslinking reactions, to improve their mechanical properties and reduce their degradation rates.

Gelatin is a naturally derived material consisting of partially hydrolyzed and degraded collagen, an essential structural protein component in the extracellular matrix of bones, cartilage and tendons. The absence of the triple helical structure in collagen, following hydrolysis and denaturation processes, results in a lower mechanical strength compared to native collagen. However, due to gelatin’s structural stability and enhanced tolerance for chemical modification, various crosslinking methods, such as physical and chemical crosslinking, are widely used to improve the mechanical properties and control the degradation of gelatin. Harnessing the presence of functional side-groups, unmodified gelatin can be chemically crosslinked using aldehydes [4], carbodiimide [5] or using a natural crosslinker such as genipin [6,7], or enzyme-mediated crosslinking can be used via transglutaminases [8,9] and tyrosinases [10]. The crosslinked gelatin hydrogels are, however, restricted, with negligible control over the degree of crosslinking and, therefore, the physicomechanical properties of the hydrogel. These limitations can be avoided by modifying the gelatin backbone with the incorporation of functional groups. Thus, through tuning the degree of functionality and crosslinking conditions, precise control over the crosslinked matrix can be achieved. One such functionalized gelatin that has been investigated in detail, for a range of applications, is gelatin methacryloyl (GelMA). The chemical modification of the gelatin backbone with methacrylic anhydride only utilizes 5% of the amino acid residues per molar ratio [11]. This implies that both cell-adhesive and matrix metalloproteinase (MMP)-sensitive motifs, which confer a wide range of biological applicability to gelatin, is retained post-modification. Furthermore, the inclusion of these photo-labile motifs in the gelatin backbone permits radical-based photopolymerization under mild conditions.

The current review focuses on recent studies related to the biomedical application of GelMA hydrogels, including their use as drug delivery systems and their antimicrobial activity, as well as their immunomodulatory properties for musculoskeletal tissue regeneration. Different search engines, such as PubMed, Web of Science and Scopus were used to search for articles published up to 31 December 2021 using the keywords “gelma” or “gelatin methacryloyl” or “gelatin” and “bone tissue engineering” or “bone regeneration” in combination with other keywords such as immunomodulation or growth factor or drug delivery or antibacterial or 3D bioprinting or nanomaterials or nanoparticles (Figure 1). Finally, we review the challenges and future development prospects of gelatin and GelMA as tissue engineering materials and drug delivery carriers.

## 2. The Synthesis of GelMA

Since the first report in 2000, GelMA has been extensively investigated to address different approaches to crosslinking the functionalized matrix for different applications where GelMA can be applied [12]. Irrespective of the numerous articles published each year, where GelMA serves as a base material, either for tissue engineering or drug/gene delivery applications (Web of Science records 286 research papers in 2021), the synthesis protocol has not deviated from the original method reported by Van den Bulcke et al. GelMA is synthesized by the chemical reaction between hydroxyl and amine groups of the amino acids residues and methacrylic anhydride (MA) [12]. Briefly, 10 wt% solution of gelatin is prepared in phosphate buffer saline (pH—7.4) at 50 °C and following complete dissolution, MA is added in excess to allow interaction of the anhydride. The reaction is continued for an hour, after which time the reaction is stopped by the addition of excess PBS (typically 5×); the solution is subsequently filtered to remove unreacted anhydride or the reaction by-product, methacrylic acid. The resulting functionalized gelatin is a mix of methacrylamide and methacrylate groups, hence the name gelatin methacryloyl [11], instead of the earlier nomenclature of gelatin methacrylate [13,14] methacrylated gelatin [15,16] or methacrylamide modified gelatin, has gained favour [17,18]. Functionalized gelatin can either be crosslinked by photopolymerization [19,20,21,22,23,24,25] or chemically crosslinked using ammonium persulfate (APS)/tetraacetylethylenediamine (TEMED) [26]. 

Irrespective of the crosslinking system used, the factors that influence the properties of the crosslinked matrix and thus its feasibility for any biomedical application are principally the degree of functionalization (a measure of methacrylation) and the macromer concentration. Independent of macromer concentration, the different degree of methacrylation (DoM) is one parameter that has been extensively investigated. The influence of the reaction conditions, such as the source of methacryloyl, the solvent used, mode of reaction, and type and source of gelatin, are known to control the DoM.

### 2.1. The Effect of Reaction Conditions (Buffer, pH and Temperature) on DoM

One strategy to modulate the DoM involves adjusting the pH of the reaction condition to prevent protonation of the primary amine. This can be achieved by maintaining the pH above the isoelectric point (pI) of gelatin (pI for type A is 8–9 whereas for type B it is in the range of 5–6) to improve the reactivity of functional groups involved in backbone functionalization. With phosphate-buffered saline (PBS) as the dissolution agent, the buffering capacity is limited and requires pH adjustments throughout the reaction duration to improve the reaction efficacy. This can be laborious and requires an excess of MA for chemical modification [27,28] and, while this is achievable, it does introduce a potential source of variability between different synthesized batches. In the absence of pH adjustments, the DoM varies in the range of 70–85%. An alternative to PBS is 0.1 M carbonate-bicarbonate buffer (CB) at pH 9.7, enabling excellent buffering above the pI of type A gelatin, maintaining the reactive amines in the neutral state. At the same molar concentration of MA, significantly different DoM is observed when the reaction is performed in PBS (51% ± 0.1) versus CB (76% ± 0.5) [29]. This can be further improved through the use of 0.25 M CB buffer with initial pH adjusted to 9 to produce a DoM of 97% ± 0.3 (Figure 2) [30,31]. Increasing the concentration of the buffer, or increasing the pH above nine, results in a decrease in the DoM due to increased hydrolysis of MA in the presence of a strong base (hydroxide ions). By increasing the molar ratio of MA/gelatin between 0.012 to 0.05, a linear increase in the degree of substitution can be achieved, providing a system where controlled functionalization can be tuned. However, this precise control over DoM is not possible when PBS is used. An important factor determining DoM is the concentration of gelatin used, where low concentrations of gelatin result in phase separation of added MA and low DoM. As the concentration of gelatin is increased, surface tension decreases thus improving MA miscibility and significantly improving DoM. In contrast, reaction temperature failed to show any effect on DoM whereas the duration of the chemical reaction showed an increase in chemical functionalization up to 30 min after reaction saturation was achieved, with no significant differences observed if the reaction was performed for 1 h versus 3 h. Kumar and colleagues investigated the role of solvent and pH together with the reaction time on the degree of hydrolysis of gelatin and modification thus identifying the optimal reaction parameter for synthesizing GelMA as a bioink for stereolithography-based (SLA) bioprinting [32]. The results confirmed the solvent-dependent effect on gelatin hydrolysis, which can be accentuated by increasing the reaction time and its downstream effect on pH and therefore DoM. This study examined the synthesis of GelMA with a slow sol-gel transition at room temperature to make it suitable for SLA bioprinting with high structural resolution. Even though the study was focused on SLA bioprinting, the work provided an explanation on the role of different reaction parameters and their effects on the physicomechanical properties of synthesized GelMA that can be easily applied to different applications where specific requirements are needed.

### 2.2. The Effect of Gelatin Source on DoM

The source of gelatin used for chemical derivatization is known to alter the properties of functionalized gelatin. Sources typically include porcine or bovine collagen, but the gelatin obtained comes with a risk of zoonosis. To overcome this issue, gelatin derived from fish has been chosen as an alternative source [33]. The source of gelatin affects the mechanical properties of hydrogels fabricated from the functionalized form which can be attributed to the number of amino acids available for modification [33,34]. For example, GelMA from a porcine source is the least viscous but displays the highest compressive modulus at room temperature when compared to gels obtained from fish-derived GelMA. Fish gelatin is suitable for the microfabrication process at temperatures below room temperature, given the low melting point of fish gelatin [35]. The nature of conditioning used for derivatizing gelatin from collagen has a negligible effect on the degree of methacrylation. Type A gelatin (GA) or acid-conditioned gelatin is used for hydrolyzing weakly interconnected collagen such as that obtained from porcine skin. Whereas, for hydrolyzing the densely interconnected collagen present in bovine skin or bovine bone, alkaline treatment is used, resulting in Type B gelatin (GB). The conditioning method results in underlying differences in gelatin properties, including amino acid composition, isoelectric point and viscosity. Irrespective of these differences, similar levels of functionalization can be achieved in both GA and GB for the same feed ratio of ml of MA/gram of gelatin. However, some differences were observed when the feed ratio of 0.05 mL MA/g of gelatin was used, where GB showed significantly higher DoM relative to GA [36,37]. Similar levels of functionalization for GA and GB result in differences including isoelectric point, viscosity, storage modulus and, particularly, the swelling ratio which decreases as the DoM increases. Understanding how these properties change with the source of material used or how DoM was achieved influences the suitability of GelMA as a bioink for different biofabrication platforms used in fabricating complex 3D constructs. 

### 2.3. Crosslinking Agents for Stabilizing GelMA Hydrogels

Like gelatin, functionalized gelatin also demonstrates a sol-gel transition where reducing the temperature results in gelation of modified gelatin that subsequently melts as the temperature increases. However, as the DoM increases, there is a decrease in the gelation and melting temperature relative to unmodified gelatin. To stabilize the fabricated hydrogel, different crosslinking agents have been used, as summarized in Table 1. Recently, studies have investigated the effect of sequential crosslinking approaches on the materials’ physicomechanical and rheological properties. Wang and colleagues reported that physical gelation of the hydrogel results in conformational changes, which, followed by photopolymerization, produced hydrogels with a high structural strength [35]. A similar approach of thermal gelation was used to bioprint GelMA physical hydrogels (GPG) at low concentrations (3 wt%) that were otherwise not possible. This relied on the ability of GPG shear thinning and self-healing properties, resulting in highly porous 3D constructs with high shape fidelity and cell viability [38]. Another study combined a controlled enzymatic crosslinking of GelMA using microbial transglutaminase followed by photo-crosslinking. The aim was to precisely tune the rheological and shear thinning properties of GelMA making it suitable for bioprinting with a focus on soft tissue bioprinting.

### 2.4. GelMA Nanocomposites 

GelMA nanocomposites are hybrid hydrogels with nanomaterials dispersed within the hydrogel matrix to improve the mechanical, rheological and biological performance of GelMA, different nanomaterials have been included and their effect investigated over years [34]. Different nanofillers have been utilized which include minerals such as hydroxyapatite, silicates and metal-based nanostructures. The inclusion of these nanostructures provides a multifunctional platform and depending on the nature of the nano-system used, enhanced hydrogel bioactivity, along with cargo delivery, can be achieved. The chemical similarity between hydroxyapatite and the mineral component of bone has made hydroxyapatite and/or calcium phosphate nanostructures one of the most extensively investigated nanocomposites. Several reports have demonstrated a degree of mechanical reinforcement of the otherwise soft hydrogel along with improved osteogenic differentiation of bone-precursor cells indicating improved bioactivity both in vitro [55,56] and in vivo [55,57,58]. Another study investigated the effect of incorporating biomimetically coated hydroxyapatite nanofibers (HANFs) with ultrahigh aspect ratio within GelMA and reported that the incorporation of 1.5 wt% of these nanostructures (15m-HANFs/GelMA) produced the best bone regeneration in vivo. The authors highlighted that even though the incorporation of 2.5 wt% of HANFs significantly improved the mechanical properties and in vitro performance of MC3T3s, it was 1.5 wt% HANFs that produced the best new bone formation in vivo. The authors suggested that the discrepancy in the in vitro and in vivo results could be related to the hydrogel network structure which could influence the nanocomposite swelling and degradation profile, thus affecting the bone regenerative effect [59]. Silicate-based nanomaterials, especially synthetic silicates such as Laponite^®^ or nano clay, have also been investigated both for improving hydrogels’ mechanical and rheological properties but also for enhancing the biological activity of biocompatible but bioinert hydrogels. Several studies have highlighted that the incorporation of these smectite nanomaterials supported osteogenic differentiation of bone-precursor cells in vitro but was also biocompatible in an immune-competent rat model in vivo [60,61]. Additionally, the nanoclays’ large and highly ionic surface area allows these nanostructures to be used as a drug/growth factor delivery system. A recent study harnessed this capability and used the laponite-GelMA nanocomposite for localized vascular endothelial growth factor delivery to stimulate vascular integration in an ex vivo chorioallantoic membrane assay. Furthermore, the nanofillers also improved the biofabrication window of GelMA bioink and supported osteogenic differentiation of stem cells under basal growth conditions [51]. There are some other nanomaterials, such as bioactive glass [62], graphene oxide [63], gold nanoparticles [64], strontium carbonate [52] and cerium oxide [65] which have also been used for leveraging the bioactivity of these nanostructures for bone tissue engineering application. The utilization of different nanomaterials for preparing nanocomposites is still in its infancy and there are several factors that need to be considered. One such factor is the capability of maximizing the utilization of these nanostructures for beneficial outcomes without causing long-term detrimental effects associated with nanomaterial toxicity. Also, efforts are being made to improve the nanofiller dispersion within the hydrogel matrix for developing composites with high filler contents and superior mechanical and biological properties [66].

## 3. Applications of GelMA with a Focus on Musculoskeletal Regeneration

### 3.1. Drug/Growth Factor Delivery

Drug delivery system strategies have been extensively employed in tissue regeneration. Gelatin-based hydrogels are recognized drug delivery carriers for various types of growth factors, antibacterial compounds and inflammatory drugs. The success of an effective drug delivery system centres on its ability to control the release of drug molecules over a defined period of time, at a defined location. For musculoskeletal tissues, which typically take longer than other tissues to repair, approaches to facilitate sustained, temporal and slow release of drugs and growth factors over the long term are crucial. Indeed, the sustained release of drugs typically improves stem cell differentiation and enhances tissue regeneration. Therefore, an array of studies has focused on functionalized gelatin-based hydrogels to release drug/s and growth factors in a slow and sustained manner [67].

The physicochemical properties of gelatin-based hydrogels are relatively easy to modify to improve polymer-drug chemical interaction and thus to control the release of drug molecules from the hydrogel network [67]. Indeed, gelatin source, charge, molecular weight and polymeric network have all been modified to maximize the drug effect (DoM modulation as detailed above) on tissue regeneration. Crosslinking the hydrogel polymeric network has been routinely applied to control the drug release profile. The drug molecules entrapped in a hydrogel network can diffuse out of the hydrogel when the polymeric network is weaker following degradation. The degradation can occur across the polymer backbone and is typically mediated by hydrolysis or enzyme activity [68,69]. Therefore, various crosslinking agents with different densities have been used to control drug release profiles via adjustment of the hydrogel degradation rate [70,71]. 

As discussed above, the poor mechanical rigidity and rapid degradation rate of gelatin hydrogels can be a challenge. Although treatment of weight-bearing bone defects typically relies on additional fixation techniques for mechanical support, maintenance of the hydrogel’s 3D structure, under local shear stresses and over the time course of repair, is critical to the hydrogel’s role as a cell scaffold and drug release system. To improve mechanical competency and degradation profile, modifying gelatin with methacryloyl followed by crosslinking with UV light radiation offers a promising approach. Crosslinked GelMA displays excellent and controllable mechanical and degradation properties as well as drug delivery efficiency [29,30]. Samorezov et al. demonstrated that BMP-2 molecules were released at a sustained rate from GelMA after an initial burst release [72]. All BMP-2 loading concentrations in the GelMA hydrogels examined showed similar cumulative release profiles. This indicates that the number of negatively charged sites on the GelMA far exceeds the number of BMP-2 molecules that can interact with the charged sites. A high crosslinking density is recognized as one of the key factors in the sustained release of drugs. Paradoxically, it has been reported that methacrylated gelatin with a high crosslinking density induced a slow degradation rate but a more rapid, burst release of drugs [73]. Interestingly, a reduction in the degree of methacrylation resulted in a significant increase in BMP-4 and basic fibroblast growth factor (bFGF) binding capacity and slow release of those molecules. Increasing methacrylation likely reduces the net positive charge that would be expected with reduced electrostatic repulsion. This highlights the importance of recognizing the competitive balance between properties that improve gel integrity and properties that sustain the binding of a drug. Thus, gel integrity and gel drug binding can have profound effects on the release of drugs from hydrogels over time.

Although GelMA has significant physicochemical properties as a drug carrier, many studies have indicated that adding other biomaterials to drugs improves the potential of GelMA synergistically. The strategy of those studies was mainly to incorporate a drug-loaded biomaterial into GelMA hydrogels to induce further sustained and slow release of the drug for bone and cartilage regeneration. Chen et al. showed that the BMP-2 release from GelMA-vascular extracellular matrix (vECM) composite hydrogels was much slower than GelMA without vECM (Figure 3A) [74]. Similarly, incorporating resveratrol (Res)-solid lipid nanoparticles (SLNs) (Res-SLNs) into GelMA demonstrated a sustained release of Res, resulting in enhanced bone formation potential [75]. Pacelli and colleagues found integrating nano-diamond-dexamethasone (ND-Dex) complex within GelMA hydrogels displayed higher retention of Dex over time, resulting in significantly increased alkaline phosphatase activity and calcium deposition [76]. Thus, in summary, drug release is typically influenced by hydrogel degradation and drug diffusion [77]. Most studies have shown that the inclusion of a second biomaterial could reinforce the hydrogel network and hinder the process of enzymatic degradation, resulting in the sustained release of drug molecules [76,78,79]. However, other groups have reported that the degradation of GelMA composite hydrogels is not relevant to the drug release profile [74,75]. For instance, although the degradation of GelMA composite hydrogels was similar to or faster than GelMA only, the drug release from the composite hydrogels was much slower than that observed from GelMA hydrogel. This suggests that the release system is more likely associated with drug diffusion in contrast to hydrogel degradation. Thus, the specific binding affinity of the drug to the second biomaterial substantially limits drug release. These systematic drug release mechanisms give prominence to a design of GelMA-based composite hydrogels that can maximize the impact of drug release.

Due to the complex process of musculoskeletal tissue regeneration, the development of dual release drug and growth factor strategies has come to the fore. A combination of angiogenesis- and osteogenesis-related growth factors are widely used to achieve vascularized bone formation. Various studies have shown that the dual release of growth factors at different kinetic rates leads to successful bone tissue repair [80,81]. To release bFGF and BMP-2 in a spatiotemporal manner, Zhou et al. designed a composite hydrogel including GelMA-bFGF and mineral and microparticles (MCM)-BMP-2 (Figure 3B) [82]. When bFGF molecules were rapidly released and BMP-2 released over time from composite gels, a significantly enhanced vascularized bone regeneration was observed compared to the single release system. The authors demonstrated that a burst release of bFGF from GelMA promoted vascularization mediated by endothelial cells, while the sustained release of BMP-2 from MCM enhanced ossification. Therefore, the dual release system synergistically enhanced new bone tissue formation. Similar to this study, Barati et al. designed a GelMA-based composite gel for spatiotemporal release of BMP-2 and VEGF [83]. Interestingly, the authors also focused on encapsulating human mesenchymal stem cells (hMSCs) and endothelial colony-forming cells (ECFCs) in hydrogels to mimic the osteoblast-vascular niche during bone development. The Timed-release of VEGF and BMP-2 significantly increased osteogenic and vasculogenic differentiation of hMSCs and ECFCs, compared to the direct addition of bFGF and BMP-2. As evidence of increased bFGF expression in the hydrogels, the authors suggested that the mineralization and vascularization may be coupled to localized secretion of paracrine signaling factors such as bFGF by the differentiating hMSCs and ECFCs. The strategy employed suggests that it is crucial to understand the complex bone regeneration process with the incorporation of the various signaling factors associated with cell functions in order to inform and drive new tissue formation.

**Figure 3 bioengineering-09-00332-f003:**
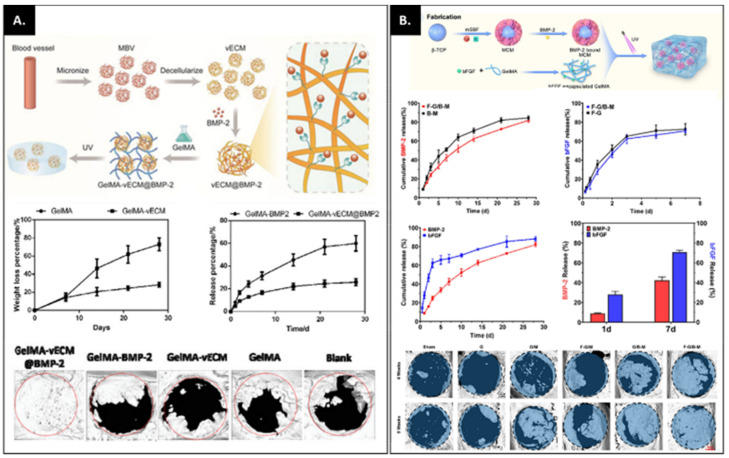
(**A**) Incorporation of vascular-derived extracellular matrix (vECM)@BMP-2 into GelMA hydrogels for angiogenic induced bone regeneration. The top panel shows a schematic diagram of GelMA-vECM@BMP-2 composite hydrogels. The middle panel represents the degradation of GelMA and GelMA-vECM hydrogels and the release patterns of BMP-2 from those hydrogels. The bottom panel shows an excellent bone regeneration ability of GelMA-vECM@BMP-2 hydrogel. Adapted with permission from Ref. [74]. Copyright 2022, Willey. (**B**) Spatiotemporal release of bFGF and BMP-2 from GelMA-based hydrogels. The top panel shows a schematic illustration of the fabrication of bFGF-GelMA (F-G)/BMP-2-MCM (B-M) composite hydrogels. The middle panel represents the release kinetics of bFGF and BMP-2. The bottom panel demonstrates that the dual release of bFGF and BMP-2 (F-G/B-M) in a spatiotemporal manner significantly enhanced bone formation compared to the single release of bFGF(F-G/M) or BMP-2 (G/B-M). Adapted with permission from Ref. [82]. Copyright 2021, Springer Nature.

### 3.2. Antimicrobial Properties (Antibiotics/Antimicrobial Compounds/Antibacterial Nanoparticles)

The nature of scaffold composition and architecture, the type, source and number of cells and the judicious addition of growth factors all play a role in developing an effective tissue regeneration strategy. A further consideration is a system with antimicrobial properties. As with any implantation strategy, biomaterials present a risk of microbial infection. Infection at the repair site impairs tissue healing and can lead to costly revision and risk of significant damage to patient health with the associated costs to economic and social well-being/quality of life of the patient and the healthcare system. In the context of bone repair, superficial bacterial colonization manifests itself into deeper infections resulting in osteomyelitis, a progressive inflammatory response leading to bone destruction [84]. The most common clinical practice to treat the infection is systemic administration of high and frequent doses of antibiotics post removal of the infected tissue which can result in drug-associated toxicity along with drug resistance. The alternative to this rather aggressive and often ineffective approach (due to poor penetration of the drug), is to design systems that can locally administer antimicrobial agents including antibiotics or other anti-infective agents thus providing a first line of defense to bacterial colonization. The ability of gelatin and/or functionalized gelatin for localized delivery of these agents in a sustained fashion over prolonged periods has been exploited in different ways. A common approach relies on combining the hydrogel with antibiotics where the drug is loaded into the matrix through passive diffusion. This was utilized by Shi et al. where colistin sulphate was loaded into the gelatin microparticles during the swelling of the dry polymer, which was then combined with polymethylmethacrylate construct. An in vitro release study confirmed the release of the antibiotic as the microparticles degraded enzymatically, releasing 10 µg/mL of antibiotic per day over the period of two weeks [85]. Similarly, gentamycin loaded into genipin-crosslinked gelatin combined with β-tricalcium phosphate scaffold provided a sustained release of the antibiotic over a period of four weeks that was effective both in vitro against *Staphylococcus aureus* and in vivo in a mouse osteomyelitis model [86]. Several different iterations were thereafter reported which combined different antibiotics with gelatin either employed as microspheres/microparticles or as bulk hydrogels that were then combined with other bioactive ceramics [87,88,89] or polymers [90,91]. Another approach relied on combining the high surface area of mesoporous silica, for sustained delivery of minocycline, with the photothermal gold nanobiopyramids (Au NBPs) dispersed within GelMA to create a hybrid system for treating periodontitis. The authors proposed that combining the bioactive minocycline-loaded silica-coated AuBNP with the hydrogel could provide localized retention and prolonged availability of the antibiotic, thereby eliminating the pathogen while the photothermal therapy could maintain reduced bacterial retention [92]. This capability is not limited to gold nanoparticles but has been expanded to other systems such as β-cyclodextrin (βCD) functionalized graphene oxide (GO) [93]. This photothermal system, combined with modified GelMA-hyaluronic acid (HA) graft, was capable of complete wound healing of bacterially infected wounds (Figure 4A). The system shows potential combination with titanium implants where the dopamine-modified HA (HA-DA) could improve metal adhesion and combination with near-infrared (NIR)-sensitive GO-βCD could target bacterial infection.

The increase in bacterial resistance to antibiotics has prompted researchers to find alternatives that are effective in addressing microbial infection. Approaches include the utilization of metal and metal nanoparticles, antimicrobial peptides (AMP) or cationic polymers. One such approach combined a short cationic AMP, HHC36, with synthetic silica nanoparticles dispersed within catechol-modified GelMA onto titanium for improved hydrogel adhesion to titanium, upregulated osseointegration and prevention of bacterial infection. The incorporation of HHC36 did not affect the physico-mechanical properties of the bioactive coating but was highly effective in the complete elimination of both Gram-positive and Gram-negative bacteria. The initial burst release of 37% in the first 24 h, followed by a sustained release of 90% of loaded AMP over the period of the next 20 days, demonstrated a promising approach to preventing implant-associated infection [94]. Metal nanoparticles (NPs) like silver [95] and zinc [96] have been combined with GelMA in different settings to impart antibacterial properties. One such study utilized functionalized GO as an active reservoir of antibacterial zinc that was combined with GelMA-phenylboronic acid to coat titanium substrates to improve cytocompatibility, and osteogenic capability, while being anti-infective against *P. aeruginosa* and *S. aureus* [96]. The hydrogel layer provides hydrophilicity to the surface, while maintaining a sustained release of the antibacterial zinc ions that was observed also to improve the cellular response. Another approach utilized a hybrid GelMA-methacrylated hyaluronic acid hydrogel as a barrier layer providing different functions [97]. Under normal conditions, the hydrogel layer controlled the release of zinc from the zinc oxide (ZnO) layer electrodeposited on titanium and permitted cell infiltration and soft tissue integration and protection against ZnO toxicity. However, under infective conditions, the hydrogel layer degrades, releasing zinc into the environment which effectively combats infection. A multifaceted hydrogel system with improved osteoimmunomodulatory and antibacterial properties was fabricated by combining silver nanoparticles and halloysite nanotubes (HNT) with GelMA (Figure 4B) [95]. The nanocomposite system was biocompatible, antibacterial and effective in supporting osteogenic differentiation under normal and inflammatory conditions in vitro and bone regeneration under infective conditions in vivo.

**Figure 4 bioengineering-09-00332-f004:**
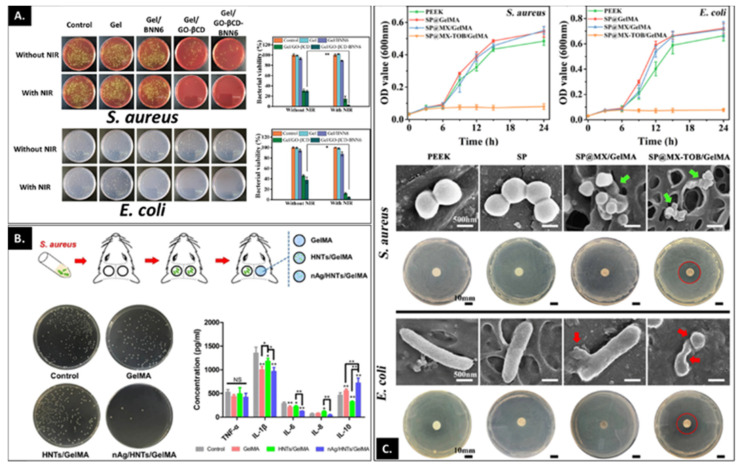
(**A**) Antibacterial activity of different substrates on agar plates and corresponding colony forming units (CFUs) against *S. au**reus* and *E. coli* with or without near infra-red radiation laser at 808 nm; laser density, 1.5 W/cm^2^ for 10 min. n = 3; error bars indicate standard deviation, * *p* < 0.05, ** *p* < 0.01. Gel: GelMA + HA-DA, Gel/BNN6: Gel containing nitric oxide donor (*N*,*N*′-Di-sec-butyl-*N*,*N*′-dinitroso-1,4-phenylenediamine), Gel/GO-βCD-BNN6: Gel containing BNN6 loaded GO-βCD Adopted with permission from Ref. [93]. Copyright 2020, American Chemical Society. (**B**) Assessing the anti-inflammatory and antibacterial properties of the nAg/HNTs/GelMA hybrid hydrogel in vivo. The panel demonstrates the schematic diagram showing the treatment of infected bone defect, bacterial colonies in the tissue of infected bone defect and the concentrations of inflammatory cytokines detected using ELISA. * *p* < 0.05, ** *p* < 0.01, NS: no statistical significance. Adapted with permission from Ref. [95]. Copyright 2020, Elsevier. (**C**) Antibacterial activity of polyetheretherketone (PEEK), SP@GelMA (GelMA coated SP), SP@MX/GelMA (MXene containing GelMA coated SP) and SP@MX-TOB/GelMA tested against *S. aureus* and *E. coli*. The top panel represents bacterial growth kinetics curves cultured on different substrates. The middle panel represents SEM images of *S. aureus* on the PEEK, SP, SP@MX/GelMA, SP@MX-TOB/GelMA for 1 day, and the inhibition zones of the corresponding substrates. The bottom panel shows morphological changes in *E. coli* cultured on varying substrates and zone of inhibition to the corresponding materials. Red arrows indicate the membrane disruption and distorted morphology of *E. coli*, green arrows indicate the *S. aureus* fragment and red circles point to the inhibition zone. Adapted with permission from Ref. [97]. Copyright 2020, American Chemical Society.

A recent study by Yin et al. [98] (Figure 4C) demonstrates an excellent example of a multifunctional platform designed for treating lesions as remnants of osteosarcoma excision that can selectively kill residual cancer cells while combating infection and supporting new bone formation. The implant utilizes sulphonated polyetheretherketone (SP) as a bioinert orthopedic implant material coated with tobramycin (TOB)-containing MXene nanosheets and GelMA (SP@MX-TOB/GelMA). Polydopamine (pDA) was used as an adhesive layer to improve interactions between MXenes and the underlying SP, which not only improved interaction between TOB-MXene and underlying PEEK, but also synergistically improved photothermal ablation of tumor cells due to MXene. TOB-MXene showed antibacterial efficacy against both Gram-positive and Gram-negative bacteria without any nephrotoxic and neurotoxic effects in vivo. The last component of this multimodal implant was GelMA which was included as a surface-modifying agent to improve implant biocompatibility and support bio-integration and new bone formation.

### 3.3. Modulation of Inflammation 

Tissue regeneration is a complex and well-orchestrated process involving three phases: inflammation, repair, and remodeling. In the inflammation phase, immune cells arrive at the defect site immediately following bone fracture. Depending on the mediators and cytokines, inflammation is resolved, followed by tissue repair and regeneration, and perturbation of this resolving inflammatory response leads to chronic inflammation. When biomaterials are implanted into the body, macrophages arrive to degrade the biomaterial and, if phagocytosis is frustrated, fuse into foreign body giant cells, in a process known as the foreign body response. Macrophages play an essential mediating role in modulating inflammation. Macrophages are often classified into M1 and M2 macrophage phenotypes. The M1 macrophage phenotype produces pro-inflammatory cytokines, such as tumor necrosis factor (TNF)-α, interleukin (IL)-6 and IL-1β. In contrast, the M2 macrophage phenotype is characterized by the production of IL-10 and transforming growth factor (TGF)-β1, essential in maintaining the long-term survival of stem and progenitor cells for tissue repair. Therefore, there has been considerable attention paid to strategies to modulate macrophage phenotype, especially from M1 to M2.

Although gelatin is a natural polymer, a foreign body response is not avoidable. After subcutaneous implantation, many infiltrated inflammatory cells have been observed in gelatin hydrogels with 0.01% or 0.1% glutaraldehyde (GA) [99]. A similar response has also been shown in non-crosslinked gelatin hydrogels. Yu et al. investigated gelatin hydrogels with increased GA crosslinking densities (0.05%, 0.1% and 0.3%) and examined the effect of crosslinking density on macrophage phenotype [100]. Interestingly, there was no effect on the macrophage phenotype and thickness of the fibrous capsule surrounding the hydrogels. 

Unlike gelatin hydrogels, GelMA reduced TNF-α production itself under lipopolysaccharide (LPS) conditions, indicating that GelMA has the potential to modulate the inflammation response [101]. Zhuang et al. demonstrated that GelMA hydrogel stiffness (2, 10 and 29 kPa) affects macrophage phenotype (Figure 5A) [102]. For example, a higher stiffness GelMA hydrogel induced greater numbers of M1 macrophages, whereas GelMA hydrogels with lower stiffness showed more M2 macrophages. This indicates that macrophage behavior is mechanically regulated by hydrogel stiffness. Interestingly, when two different hydrogels, GelMA and poly(ethylene glycol) diacrylate (PEGDA), have a similar stiffness (3.0 ± 0.3 kPa), GelMA showed significantly lower M1 marker expression and higher M2 expression compared to PEGDA (Figure 5B) [103]. Furthermore, when IL-4, the most effective cytokine at polarizing M1 phenotype to M2, was incorporated into the GelMA, the hydrogel dramatically down-regulated M1 related genes and upregulated M2 genes, compared to the PEGDA with IL-4. These findings demonstrate that GelMA has excellent potential for suppressing pro-inflammatory cytokines and promoting anti-inflammatory cytokines. Nevertheless, to date, only a handful of studies on the immunomodulatory effect of GelMA through drug delivery or application in 3D printing have been reported. With increased attention to immunomodulation and macrophage polarization for bone regeneration [104], we expect that GelMA will be widely used to modulate inflammation response for musculoskeletal regeneration.

### 3.4. 3D Bioprinting and GelMA as a Cell Delivery Platform

To design tissue-mimicking constructs with defined architecture, 3D printing, also known as additive manufacture, has gained significant interest as an approach by which spatial and temporal control of different layers can be achieved [105,106]. This opens the avenue of preparing complicated constructs with defined zonal distribution of the same or different material and cell types that are not possible by casting or molding a cell-laden material. The ease of synthesis, biocompatibility and the ability to tune chemical functionality has made GelMA one of the most frequently investigated materials as a bioink for manufacturing 3D constructs for hard and soft tissue regeneration. In particular, for bone regeneration applications, bioprinting entails the use of GelMA as a cell-laden bioink with a scaffolding structure as a mechanical backbone [32,107,108,109]. The encapsulation of the cells within the hydrogel matrix provides a transient native tissue mimicking microenvironment for the cells to interact with a mechanically compatible scaffold instead of relying on cell-scaffold interaction via cell adhesion [110]. The printed constructs are then implanted directly post-fabrication or in some cases are cultured in vitro to allow tissue maturation before implantation in vivo. 

Different printing platforms have been utilized for bioprinting 3D GelMA constructs with high complexity utilizing computer-aided designs which rely on either extrusion-based bioprinting (EBB) or non-extrusion (inkjet or lithography assisted)-based platforms. Each platform has its own advantages and limitations. For example, while inkjet printing is relatively fast and has high resolution, it is also limited in the thicknesses that can be achieved and the cell density able to be incorporated. On the other hand, EBB permits the utilization of high cell densities but is often restricted by the resolution of the printed construct (200–1000 µm). The cost-effectiveness, ease of accessibility and ability to combine different materials (thermoplastics, hydrogels and/or growth factors and cells) with spatial organization has made EBB the most investigated printing platform. Irrespective of the printing platform selected, GelMA has been most extensively utilized as bioink/bio-resin after taking rheological and technical constraints into consideration. 

In this review, we focus primarily on extrusion-based bioprinting where GelMA is used as a base material and combined with other polymers or nano-fillers to achieve different functional attributes in the printed construct. The ability of the bioink to be utilized for extrusion-based printing relies on the ability of the ink to resist flow before extrusion but when high force is applied, using air pressure or through a piston, the viscosity reduces to permit continuous filament flow through the nozzle. The extruded filament thus stacks on top of the previously printed layer and retains shape during crosslinking. Printing soft materials such as GelMA with low hydrogel network densities poses significant challenges that can be overcome either by using a template sacrificial material such as pluronics or by significantly reducing the printing temperature conditions to form a stable extruded filament. Alternatively, the inclusion of nano-fillers (Figure 6A,B) and/or other polymers (synthetic or natural (Figure 6C,D)) entail improved print fidelity by altering their rheological properties and biological functionality. The use of additives such as polymers started as a passive approach to improve the rheological properties of the hydrogel and permit printability of materials under fragile conditions. However, there has been a gradual shift in the focus where designing multifunctional 3D-printed constructs is desirable. The desired output not only improves bioink printability and shape fidelity of multi-layered printed constructs but also enhances the viability and functionality of cell-laden scaffolds. Table 2 summarizes a list of polymeric and bioactive additives used for bioprinting GelMA with a focus on bone regeneration.

Irrespective of the additives used to improve the physicomechanical properties, printability and biological functionality of GelMA for bone regeneration, these materials need additional mechanical reinforcement for designing 3D patient-specific constructs with appropriate physicomechanical requirements. ‘Hybrid biofabrication’ is one of the strategies employed and involves designing a multi-material construct combining bioink as a cell-instructive component with biomaterial ink as a structural scaffold. This allows combining components with appropriate cell-specific cues to permit cell viability and functionality whereas ‘biomaterial ink’ improves its mechanical strength along with topological cues to dictate cell function. One popular approach centres on drop casting the pre-polymerized cell-laden GelMA onto a scaffolding construct. A recent study by Qiao et al. prepared an osteochondral construct mimicking the structural and compositional gradient of the native tissue [111]. This was achieved by combining a polycaprolactone (PCL) scaffold prepared by melt “electro-writing” with varying fiber organization with MSC-laden GelMA with TGFβ1 and/or BMP7 or BMP2-containing PLGA microspheres. This enabled recapitulation of the superficial, deep and bone region of the osteochondral construct thus resulting in differential cellular phenotype and ECM deposition in vitro along with subchondral bone and cartilage regeneration in vivo. The focus of the study was on osteochondral regeneration, but similar strategies can be employed for different tissues where, by mimicking the structural and cellular composition of the tissue of interest, tissue regeneration can be achieved. Another strategy employed printing a hydrogel-engraved PCL scaffold where a layer of printed PCL was engraved using a 22 G needle at high temperature and the void created was then filled with low viscosity GelMA via a 27 G needle [112]. In this study, the high porosity of the printed construct was maintained which allowed the combining of cell-laden soft hydrogels, which could otherwise not be printed. The layer-by-layer approach proposes a strategy where precise localization of cell-type and/or growth factors laden hydrogel can be achieved within the engraved construct without compromising the material porosity, connectivity and mechanical properties. However, the study was limited by the amount of hydrogel material that could be combined with the polymer matrix, but once optimized, the material can be used to design complex multi-cellular constructs with controlled spatial presentation of cellular and bioactive factors for designing tissue-engineered constructs.

**Figure 6 bioengineering-09-00332-f006:**
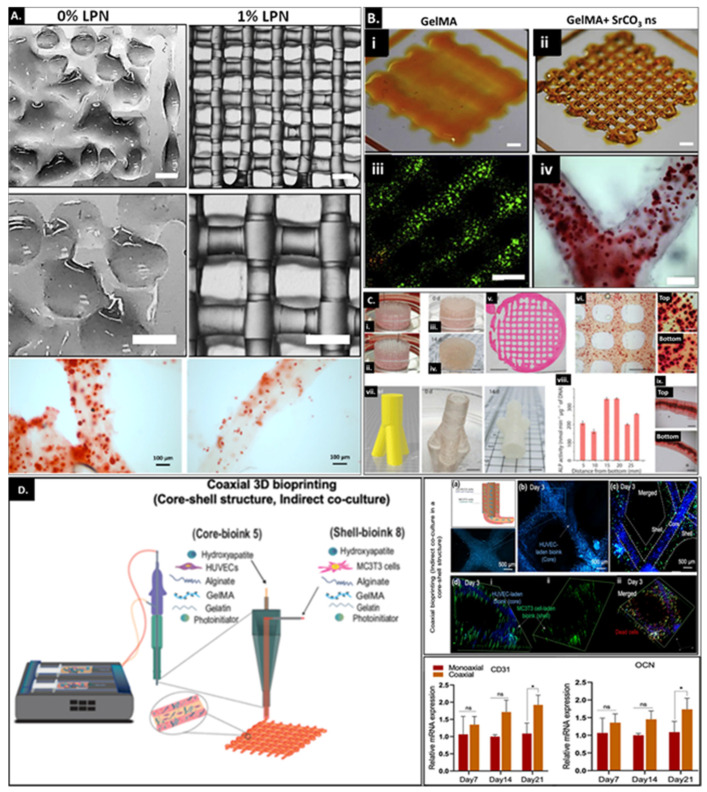
(**A**) Comparison of printing fidelity achieved by 7.5 wt% GelMA and 7.5 wt% nanocomposite bioinks incorporating 1 wt% LPN (Scale bars: 1 mm); low magnification (top panel) and high magnification (middle panel); bottom panel represents matrix mineralization of HBMSCs-laden 3D-printed LPN-GelMA construct cultured under osteogenic differentiation conditions (left) and media without dexamethasone (right) for 21 days. Adapted with permission from Ref. [51]. Copyright 2019, IOP. (**B**) Comparing printability of GelMA (i) with GelMA incorporating SrCO_3_ nanostructures (ii) which also supports high cell viability of encapsulated MSCs as determined by live (green)/dead (red) staining (iii) matrix mineralization assessed via Alizarin Red staining of mineralized (CaP) nodules (iv) in bioprinted cell-laden Sr-GelMA scaffold. Adapted with permission from Ref. [52]. Copyright 2020, Elsevier. (**C**) GelMA+ (5 wt% GelMA with 5% gelatin crosslinked using LAP) bioprinted to form a porous cylinder construct (diameter, 2 cm; height, 1 cm) with Saos-2 cells in the bioink before (0 day) (i and iii) and after culture (14 days) (ii and iv). (v) The microscopy image (hematoxylin and eosin staining) of the cylinder cross-section indicates the maintenance of a uniform pattern during tissue formation and (vi) Alizarin Red S staining of printed samples (14 days). The higher-magnification images indicate the top and bottom layers along the height of the cylinder. (vii) Model used for fabricating trifurcated tubular bioprinted constructs and the final printed construct. (viii) normalized ALP activity and (ix) Alizarin Red S staining of samples at different positions along the length of the trifurcated tube after 14 days of culture. GelMA+ (5 wt%), Saos-2 (7.5 × 10^6^ cells/mL) and osteogenic medium were used throughout. Adapted with permission from Ref. [113], Copyright 2020, Creative Commons CC-BY-4.0 license. (**D**) Left: schematic illustration of coaxial bioprinting techniques using human umbilical vascular endothelial cell (HUVEC)-laden angiogenic bioink (core-bioink) and MC3T3-laden osteogenic bioink (shell bioink) printed via the core and shell nozzle, respectively. Right top: confocal fluorescence micrograph of the core-shell structure on day 3 of culture. HUVECs (encapsulated in the core bioink) were in the center of the filament surrounded by MCT3T3 cells in the shell. HUVECs were labeled in blue using ER-cell tracker, MC3T3 cells were stained in green using Calcein-AM and dead cells were stained in red using ethidium homodimer. Right bottom: relative gene expression analysis of CD31 (angiogenesis) and osteocalcin (osteogenesis) in HUVECs and MC3T3 cells encapsulated in the homogeneous (direct co-culture) and core-shell (indirect co-culture) structures printed using monoaxial or coaxial bioprinting techniques, respectively, on days 7, 14 and 21 of culture. Data are presented as mean values ± standard deviations (n = 4). Significant differences are shown with * *p* < 0.05, ** *p* < 0.01, and ns indicates the nonsignificant differences. Adapted with permission from Ref. [114]. Copyright 2022, Wiley.

Contrary to the strategies mentioned earlier, a recent study focused on multi-material extrusion printing to design a hybrid bone-specific construct utilizing PCL to provide structural integrity and mechanical strength to the printed construct where GelMA served as a reservoir for stem cells [53]. Combining PCL with magnesium hydroxide nanostructures as bioactive nanofillers demonstrated improved degradability under accelerated in vitro degradation conditions. These nanostructures enhanced the mechanical properties of solid and porous scaffolds but also improved materials’ biological functionality without compromising their printability and shape fidelity. The bioink on the other hand included strontium carbonate nanostructures previously reported to improve the printability and osteogenic capacity of GelMA. Relative to the hybrid construct fabricated using PCL with Sr-GelMA, Mg-PCL combined with Sr-GelMA significantly upregulated osteogenic differentiation of encapsulated MSCs demonstrating functional synergies between different components of the hybrid construct. This strategy represents another approach to combining different materials with varied bioactive cues (varied spatial distribution of different materials and/or cells/growth factors) which can work in harmony to support the development of complex multifunctional constructs for tissue regeneration. 

**Table 2 bioengineering-09-00332-t002:** Polymeric and bioactive additives for bioprinting GelMA for bone regeneration.

Additive	Cell Source	Photoinitiator Used for Crosslinking	Key Finding	Ref
Gelatin (Figure 6C)	Saos2 (human osteosarcoma cell line)	LAP	Inclusion of 5 wt% gelatin in 5 wt% GelMA to form a complementary bioink permits printability of complex structures. These include printing a bone-like geometry which was 4 cm long, 2 cm wide and 1 cm high and a 3 cm high and 1.5 cm wide trifurcated tube with hollow interior and overhanging walls. The different printed constructs displayed the same levels of ALP activity and matrix mineralization in different segments of the construct.	[113]
Gelatin, alginate and hydroxyapatite	MC3T3 (mouse pre-osteoblast cell line) and HUVECs (Human umbilical vein endothelial cells)	Irgacure I-2959	Co-axial printing results in a 3D-printed construct with a core-shell structure with endothelial cells-laden ink forming the core and the MC3T3-laden ink forming the shell of the extruded fiber. Significant upregulation in osteogenic and angiogenic activity was observed for the osteon-like structures relative to the constructs printed via monoaxial 3D bioprinting.	[114]
Gelatin microgel (Figure 6D)	MC3T3/HUVECs	LAP	Combining sacrificial gelatin microgels with GelMA allows development of printed constructs with mesoscale pore networks for enhanced nutrient delivery and cell growth. The encapsulated cells demonstrate improved bioactivity within printed constructs ≥1 cm. The effect of the mesoscale porosity on cell functionality and tissue maturation still needs investigation.	[115]
Gellan gum (GG) and polylactic acid (PLA) microparticles as stem cell carriers	Rat MSCs	Irgacure I-2959	Microcarrier MSCs (MCs) complexes were formed by utilizing PLA-based particles with MSCs adhered to their surface. The MCs containing GelMA-gellan gum bioink formed the bone compartment of the osteochondral construct. The inclusion of MCs provided mechanical reinforcement to the construct, whereas incorporation of GG improved viscosity and printability of the bioink.	[42]
Hydroxyapatite (HAp) and methacrylated hyaluronic acid (HAM)	hASCs (human adipose-derived stem cells)	LAP	HAp ink was prepared by incorporating HAp (5 wt%) within gelatin methacryloyl of different degrees of methacrylation and hyaluronic acid (7 wt% GM2, 5 wt% GM5 and 1 wt% HAM). HAp bioink demonstrated improved printability with printed structures remaining structurally intact over a 28-day period. Furthermore, the inclusion of HAp showed an osteo-supportive effect with upregulated osteogenic differentiation and matrix mineralization in osteogenic and control culture conditions.	[116]
Gelatin (G), acetylated gelatin methacryloyl (GMA), hydroxyapatite (HAp) and methacrylated hyaluronic acid (HAM)	ASCs (adipose derived stem cells) and HDMECs (human dermal microvascular endothelial cells)	LAP	Inclusion of GMA and G within GM for preparing the vascular bioink allowed improved materials properties with reduced crosslinking density and high swelling which allows capillary formation and maintenance. The combination of the vascularized bioink with the bone bioink (G, GM, HAP and HAM) demonstrated formation of a stable capillary-like network along with improved expression of bone-matrix-specific proteins relative to monoculture controls.	[117]
Gelatin, polyethylene glycol and mesoporous calcium silicate nanostructure	rBMSCs (rat bone marrow stem cells) and RAW264.7	LAP	Incorporation of 3% gelatin, 2% PEG and 0.4% MSN within 5% GelMA improved hydrogel physicomechanical properties and bioink printability. Additionally, inclusion of BMP4-loaded MSN supported M2 type polarization, osteogenic differentiation of rBMSCs in vitro and accelerated bone healing in the critical-sized calvarial defect in a diabetic mouse model.	[118]
Bone Particles (BP)	Cells native to BP	LAP	Inclusion of BP with 0–500 µm size distribution within 10% and 12.5% GelMA at the filler concentration of 15% *w*/*v* improved bioink printability and mechanical properties. Additionally, the cellular reserve from the viable BPs displayed cell migration and colonization of the hydrogel scaffold while retaining their osteogenic differentiation capability relative to scaffolds with BP in the size range of 150–500 µm.	[119]
Mesoporous silica nanoparticles (MSN) functionalized with calcium phosphate (CaP) and dexamethasone (Dex) (MSNCaPDex)	Human MSCs	Irgacure I-2959	Inclusion of MSNCaPDex at 0.5% *w*/*v* concentration within 10% GelMA improved hydrogel printability and shape fidelity while supporting stem cell viability and osteogenic differentiation capability in the basal media without additional osteogenic factors included during culture conditions.	[120]
Laponite^®^XLG (Figure 6A)	Human MSCs	Ru/SPS	Inclusion of Laponite served multifold functionality where Laponite served as a growth factor reservoir, improved bioink printability and promoted osteogenic differentiation capability of encapsulated stem cells along with integration and vascularization of the implanted construct in the chick chorioallantoic membrane model.	[51]
SrCO**_3_** (Figure 6B)	Human MSCs	Ru/SPS	Utilization of SrCO**_3_** as a nanofiller within 5 wt% GelMA improved printability and shape fidelity of the printed construct over prolonged culture periods and enhanced osteogenic differentiation of encapsulated stem cells.	[52]

## 4. Concluding Remarks and Future Perspectives

GelMA has been recognized as one of the most promising hydrogel platforms with widespread applicability for 3D bioprinting and tissue regeneration. This is attributed to GelMA’s biocompatibility, ease of synthesis in a laboratory setting and the ability to tune the physicomechanical properties of the crosslinked matrix by changing the degree of modification. The similarity to the ECM and feasibility of incorporating bioactive factors such as antimicrobial agents, growth factors etc. has prompted its utilization in tissue regeneration, drug delivery and as tissue sealants. In addition, efforts have been expended to prepare GelMA-based hybrid constructs by combining GelMA with other materials such as synthetic or natural polymers and inorganic or organic nanomaterials. The resulting hybrid material leverages the advantages that the individual components have to offer while improving the mechanical and rheological properties and/or bioactivity of the composite. Another research avenue where GelMA has been extensively investigated is 3D bioprinting whereby changing the composition, inclusion of additives and the platform used for printing has been tried to fabricate 3D constructs of varying shapes and over varying length scales. From designing simple lattice structures to test the printability of the ink to fabricating hierarchical structures with precise spatio-temporal distribution of cells and/or bioactive factors has been the focus of research over the last few years. To design complex, patient-specific constructs with retained bioactivity and structural stability, hybrid biofabrication strategies are currently under investigation. The hybrid biofabrication strategies involve combining cell-laden GelMA bioink with thermoplastic polymers as biomaterial ink printed in a layer-by-layer fashion to fabricate a hybrid construct with high mechanical strength and improved bioactivity. An alternative approach will be modular fabrication, where cell-laden/growth-factor laden GelMA are prepared as bioactive modules that can then be uniquely positioned in a printed construct. 

Designing complex hybrid constructs with anatomical specificity and tissue complexity is still relatively new and efforts are in progress to combine different materials and cells using a variety of new manufacturing techniques. This will therefore permit the printing of multifunctional hybrid constructs that are mechanically matched to the tissue of interest with a structure that permits tissue integration. However, developments and improvements await in terms of (i) improved fine-tuning of the biodegradability profile of GelMA-based constructs to adapt to the specific healing and regenerative needs of each musculoskeletal tissue, (ii) better understanding of the bioinstructive properties of GelMA itself to facilitate translation, (iii) GelMa derivatization with motifs that would provide more precise, (iv) new modes of spatial-temporal control of the bioactive cues and reduction of printing time for these constructs, and (v) a better understanding of the potential for printed constructs to be directly implanted into the patient or whether the constructs will need additional in vitro maturation. As these new developments and improvements for the use of GelMA-based constructs arise, further advancements should be made to regulate and approve GelMA-based biomedical products to help clinicians and patients in the near future.

## 5. Summary

GelMA has the advantage of holding both natural and synthetic polymer properties (e.g., good mechanical properties, biocompatibility and easy chemical modification). Compared to other tissues, musculoskeletal tissues are additionally required for mechanical support and biological stimulation over the life course The tuneable physicochemical and mechanical properties of GelMA hydrogels offer exciting approaches to the incorporation of cells and bioactive molecules (e.g., growth factors, antibiotics and drugs) and in combination with other biomaterials, to aid musculoskeletal tissue regeneration.

The future is bright for reparative applications using GelMA with new opportunities to advance musculoskeletal healthcare for an increasingly ageing demographic. The challenge will be the understanding of the design of GelMA-based complex hybrid constructs that can orchestrate musculoskeletal tissue regeneration. The key will be studying to understand the cellular mechanism, including macrophage-associated inflammation modulation and stem cell-associated tissue remodeling. Given the excellent biocompatibility and physicochemical and mechanical properties of GelMA, we anticipate new approaches that will enable mimicking of native tissue structure and stimulation of the cell reparative niche to generate new tissue. The acceptance of the fundamental importance of the complex musculoskeletal tissue structure and process of reparation therein, offer a springboard to new approaches to using GelMA for tissue regeneration.

## Figures and Tables

**Figure 1 bioengineering-09-00332-f001:**
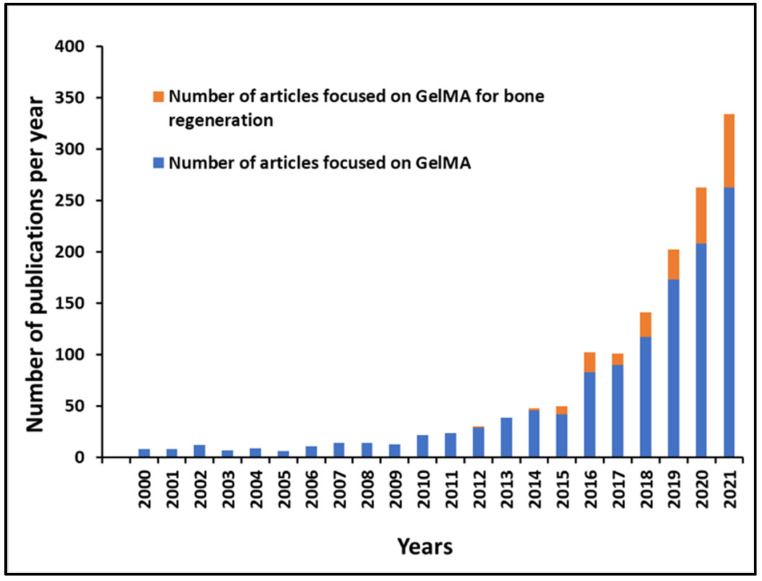
Number of articles focused on GelMA and GelMA for bone regeneration published per year since 2000 according to PubMed.

**Figure 2 bioengineering-09-00332-f002:**
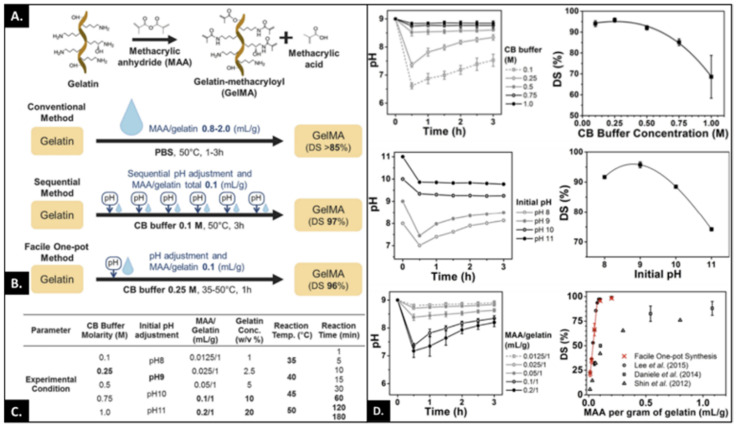
(**A**) Schematic illustration of GelMA synthesis. (**B**) represents different synthesis processes adapted with their corresponding DS (degree of substitution). The conventional method involves adding a large amount of methacrylic anhydride (MAA), but sequential addition requires less amount of MAA and pH adjustments to achieve higher DS. The other method relies on one-pot synthesis where high DS is achieved. (**C**) represents different experimental conditions tested and the corresponding reaction times. Letters/numbers in bold represent optimum conditions. (**D**) The graphs on the left represent pH changes during reaction conditions whereas the graphs on left represent DS as a function of molar concentration of CB (carbonate buffer), initial pH and MAA/gelatin ratio. Error bars represent the relative standard deviation of n = 3. Adapted with permission from Ref. [30]. Copyright 2016, Springer Nature.

**Figure 5 bioengineering-09-00332-f005:**
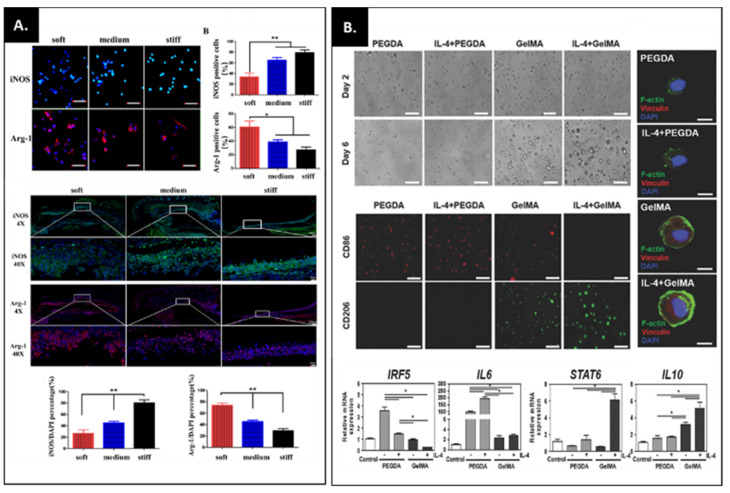
(**A**) Macrophage marker expression on GelMA hydrogels with different stiffnesses (iNOS; M1 marker, Arg-1; M2 marker). This study demonstrated that the soft GelMA hydrogel enhanced M2 polarization in vitro and in vivo. Adapted with permission from Ref. [102]. Copyright 2020, American Chemical Society. (**B**) Assessing macrophage morphology and phenotype in IL-4 incorporated PEGDA and GelMA hydrogels. The macrophages in PEGDA hydrogels showed clump-like cytoplasmic aggregates of F-actin, whereas macrophages in GelMA showed the presence of a prominent cortical shell. The staining images with M1 surface marker (CD86) and M2 surface marker (CD206) and qPCR of M1-related KRF5 and IL-6 and M2-related STAT6 and IL-10 indicate that the GelMA hydrogel with IL-4 enhanced M2 polarization, compared to the PEDGA-IL hydrogel. Adapted with permission from Ref. [103]. Copyright 2017, WILEY-VCH Verlag GmbH & Co. KGaA, Weinheim. * *p* < 0.05, ** *p* < 0.01.

**Table 1 bioengineering-09-00332-t001:** Different crosslinking systems used for polymerizing GelMA.

Cross-Linking System	Biological Response	Ref.
APS/TEMED	Encapsulated chondrocytes showed >80% viability after 24 h.	[26]
Eosin Y (photosensitizer), Triethanolamine (TEA; initiator) and Vinylcaprolactam (VC; co-monomers)	Viability both in 2D and 3D cultures is dependent on hydrogel formulation (concentration of macromer, Eosin Y, TEA and VC and crosslinking time) along with in vivo biocompatibility and bone-forming capability.	[39,40,41]
Irgacure I-2959	Cell viability was dependent on the concentration of Irgacure and duration of crosslinking. The system has been extensively investigated in the literature. however there is a gradual drift towards crosslinking systems using visible light due to the associated negative effect on the cytotoxicity and cell functionality with the UV-light source.	[16,25,42,43,44,45,46,47]
Lithium phenyl-2 4 6-trimethylbenzoylphosphinate (LAP)	Cell viability of >75% which is dependent on crosslinking conditions including macromer concentration, LAP concentration and time of crosslinking; good cytocompatibility especially at high photo-initiator concentrations (0.7% *w*/*v*) during prolonged bioprinting conditions (60 min) with small pore size and low swelling ratio and slower degradation.	[48,49,50]
Ruthenium/sodium persulfate (Ru/SPS)	Superior cell viability (>80% over long-term cultures) and support cell differentiation capabilities (osteogenesis, chondrogenesis).	[25,51,52,53]
Riboflavin	Improved viability and expression of late osteogenic markers such as osteocalcin of KUSA-1 (murine bone marrow-derived MSCs committed towards osteocyte differentiation) in 20% GelMA crosslinked with riboflavin relative to hydrogels crosslinked using Irgacure I-2959.	[54]

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
