# Peer review of "Gelatin Methacryloyl Hydrogels for Musculoskeletal Tissue Regeneration"

_bioengineering, 2022, doi:10.3390/bioengineering9070332_

Round 1
Reviewer 1 Report
Kim et al describe the recent progress in hydrogels more broadly and the highlight recent advances of GelMA hydrogels in regards of the properties and application in the emerging fields of musculoskeletal drug delivery and material design. The review concludes with future perspectives and associated challenges for developing local drug delivery for musculoskeletal applications. The review is well structured; moreover, the topic is of interest and up to date.
The results are proper for publishing in Bioengineering, however, there are issues that must be solved before its recommendation for publication, such as:
- The authors didn’t explained if they did a systematic work to identify the papers of interest for the present review, also if they did use a specific selection methodology. Tables consisting of the articles of interest are added but, how you select those studies, what criteria did you considered? Are these studies the only ones of interest? I suggest to use Pubmed, Scopus or other search engine to detect all the studies of interest in recent years. A relevant graph with the number of the studies conducted in recent years on GelMA containing materials, would be helpful for statistic data.
- Please, include Figure 2.
- Some editing problems are present e.g line 421 Tthe; line 488-3D printing; line 455 [88]. please, revise the whole manuscript.
- please, include a special paragraph regarding GelMA nanocomposites (with clay, silica, metallic particles) as these are of interest for muscoskeletal applications.
I would suggest publishing the manuscript after minor revision is performed.
With respect,
Author Response
Reviewer 1:
We are grateful to reviewer 1 for the constructive remarks on the review. We have incorporated all the suggested and requested changes in the revised manuscript.
The results are proper for publishing in Bioengineering, however, there are issues that must be solved before its recommendation for publication, such as: The authors didn’t explained if they did a systematic work to identify the papers of interest for the present review, also if they did use a specific selection methodology. Tables consisting of the articles of interest are added but, how you select those studies, what criteria did you considered? Are these studies the only ones of interest? I suggest to use Pubmed, Scopus or other search engine to detect all the studies of interest in recent years.
A relevant graph with the number of the studies conducted in recent years on GelMA containing materials, would be helpful for statistic data.
We thank the reviewer for these useful points and suggestions. Please note we have addressed the points throughout in the revised script - We added Figure 1 detailing the number of studies focussed on GelMA and GelMa for bone regeneration.
Please, include Figure 2. Some editing problems are present e.g line 421; line 488-3D printing; line 455 [88]. please, revise the whole manuscript.
We thank the reviewer for these points and have corrected all items raised. In the revised manuscript.
Please, include a special paragraph regarding GelMA nanocomposites (with clay, silica, metallic particles) as these are of interest for muscoskeletal applications.
We thank the reviewer for this constructive suggestion - Please note we have added a whole new section: “2.4 GelMA Nanocomposites” to describe GelMA nanocomposite applications.

Reviewer 2 Report
In this review, the authors present knowledge about GelMA and their applications in drug delivery and regenerative medicine. This review is very comprehensive and informative. Following details would improve the manuscript:
1. This manuscript includes a mix of British English and American English. Should be unified.
2. Table 1: In this table, the item for the cross-linking system using Eosin Y, Triethanolamine, and Vinylcaprolactam is overlapped.
3. Figure 2 is missing in the text.
4. Words with the same meaning should be unified. For example, “tuneable” and “tunable”.
Author Response
Reviewer 2
In this review, the authors present knowledge about GelMA and their applications in drug delivery and regenerative medicine. This review is very comprehensive and informative. Following details would improve the manuscript:
- This manuscript includes a mix of British English and American English Should be unified.
We thank reviewer 2 for the constructive comments on the review – we, as requested, unified the text to American English.
- Table 1: In this table, the item for the cross-linking system using Eosin Y, Triethanolamine, and Vinylcaprolactam is overlapped.
We thank reviewer 2 for the observation. Please note we have removed the overlapped row.
- Figure 2 is missing in the text.
We thank reviewer 2 for identifying this omission- we have added Figure 2 to the text.
- Words with the same meaning should be unified. For example, “tuneable” and “tunable”.
We thank reviewer 2 for this observation- as requested, we have unified the text to American English and stated as tuneable.
Reviewer 3 Report
THe paper is a ewell written review on gelatin methacryloyl 26 (GelMA) hydrogel and its use as delivering systems or bioinks.
The review includes many updated relevant data and discussed the litterature in a balanced way.
Only one minor concern: genus and species of bacteria must by italicized
Author Response
Reviewer 3
The paper is well written review on gelatin methacryloyl 26 (GelMA) hydrogel and its use as delivering systems or bioinks.
The review includes many updated relevant data and discussed the literature in a balanced way.
Only one minor concern: genus and species of bacteria must by italicized
We are grateful to reviewer 3 for the constructive remarks on the review. We thank reviewer 3 for the observation that genus and species of bacteria must be italicized - please note we have now as requested.